

# Robust 3D object watermarking scheme using shape features for copyright protection

Sarah M. Alhammad[1], Nada Ahmed[1], Safia Abbas[2], Hussein Abulkasim[3,4] and Ahmed Elhadad[5]

[1] Department of Computer Sciences, College of Computer and Information Sciences, Princess Nourah bint Abdulrahman University, Riyadh, Saudi Arabia
[2] Computer Science Department, Faculty of Computer and Information Sciences, Ain Shams University, Cairo, Egypt
[3] Department of Information Technology, University of Science and Technology of Fujairah, Fujairah, UAE
[4] Faculty of Science, New Valley University, El-Kharga, Egypt
[5] Department of Computer Science, Faculty of Computers and Information, South Valley University, Qena, Egypt

## ABSTRACT

This article utilizes the discrete wavelet transformation to introduce an advanced 3D object watermarking model depending on the characteristics of the object's vertices. The model entails two different phases: integration and extraction. In the integration phase, a novel technique is proposed, which embeds the secret grayscale image three times using both the encrypted pixels and the vertices' coefficients of the original 3D object. In the extraction phase, the secret image is randomly extracted and recaptured using the inverse phase of the integration technique. Four common 3D objects (Stanford bunny, horse, cat figurine, and angel), with different faces and different vertices, are used in this model as a dataset. The performance of the proposed technique is evaluated using different metrics to show its superiority in terms of execution time and imperceptibility. The results demonstrated that the proposed method achieved high imperceptibility and transparency with low distortion. Moreover, the extracted secret grayscale image perfectly matched the original watermark with a structural similarity index of 1 for all testing models.

Corresponding author
Safia Abbas,
safia_abbas@cis.asu.edu.eg

## INTRODUCTION

In today's world, online browsing has become an integral part of our lives. However, such an online open environment embraces multifarious duplications of digital data and objects, creating multiple illegal copies from the source object (*Haynes, 2022*). Copyright laws protect various forms of digital creation, including e-books (*Lauwda, Gemilang & Ferguson, 2023*; *Chou et al., 2021*), images (*Wan et al., 2022*; *Hsu, Hu & Chou, 2022*; *Hamad, Khalifa & Elhadad, 2014*), videos (*Asikuzzaman & Pickering, 2017*), music (*Czerwinski, Fromm & Hodes, 2007*; *Huang, 2023*), databases (*Brown, Bryan & Conley, 1999*), and 3D objects (*Al-Saadi, Elhadad & Ghareeb, 2021*). Such laws hinder users from

coping with digital sources and provide intellectual property protection to the main owners. In the case of digital media, copyright gives the owner exclusive legal rights to have copies of their authentic works. With the rapid development of digital technologies, 3D objects have become increasingly popular and commonly used in different domains such as entertainment, education, and manufacturing (*Chuvikov et al., 2014*). Despite the use of 3D objects becoming more prevalent, there is a growing concern over intellectual property rights and copyright protection. Unauthorized use and distribution of 3D objects can lead to significant financial losses for the creators and owners of these objects. Accordingly, it is mandatory to develop practical models for protecting the intellectual property rights of 3D objects, such as watermarking (*Al-Saadi, Ghareeb & Elhadad, 2021*).

In response to this need, the watermarking model has been considered a promising solution for copyright protection, especially for 3D objects. By embedding a unique signature or identifier within the object, watermarking can help deter unauthorized use and distribution and enable copyright owners to track the use of their content. However, developing effective watermarking techniques for 3D objects poses several challenges, such as ensuring the watermark is robust to various attacks while maintaining its invisibility to the human eye (*Kumar, Singh & Yadav, 2020*; *Wan et al., 2022*; *Kumar et al., 2023*). Recently, several watermarking methods have been presented for 3D objects, but there is still a need for more robust and efficient methods. Many existing techniques suffer from limitations such as low robustness, low capacity, and low invisibility, which can make them vulnerable to attacks and unauthorized removal of the watermark (*Medimegh, Belaid & Werghi, 2015*).

One of the main challenges in watermarking solutions, especially for 3D objects, is the integrity maintenance of 3D objects while embedding the watermark. Since 3D objects consist of complex structures and details, any modification to the object can potentially affect its visual quality and functionality (*Dugelay, Baskurt & Daoudi, 2008*). Therefore, developing an advanced watermarking model that embeds the watermark without significantly altering the original object is important. This requires a careful balance between the strength of the watermark and the object's visual quality. Moreover, in 3D object watermarking, it is important to ensure the security of the embedded watermark. In other words, it is essential not to permit unauthorized users to remove or update the watermark with no guarantee of permission from the main owner. Such security measures can be achieved by encryption and digital signatures, which protect the watermark from manipulation and unauthorized access (*Zhou et al., 2023*; *Yeo & Yeung, 1999*). Given the limitations of traditional copyright protection methods for 3D objects, a robust watermarking model has been proposed.

This work offers a new 3D object watermarking-based paradigm that seeks to robustly and covertly implant a secret message within the item. Our suggested model involves embedding a grayscale image three times and using the DWT of the 3D object vertices. The experiment used four 3D objects (Stanford bunny, horse, cat figurine, and angel) with different faces and vertices, as seen in 'Experimental Results'. We evaluate the performance of our method by measuring various aspects and comparing the results of the 3D object before and after the watermarking process. The research findings indicate that the suggested

model achieved better performance regarding execution time and invisibility, making it a promising solution for protecting 3D object copyright.

Section 'Related Work' reviews the related work to provide context for our approach, and 'Methodology' presents the details of our watermarking model, including the embedding processes. The experimental results and analysis are introduced in 'Experimental Results', including a comparison between the performance of the original 3D object and the watermarked version. Finally, the summarization and findings are presented as a conclusion in 'Conclusion'.

## RELATED WORK

Watermarking techniques for 3D objects have attracted many researchers in recent years, with numerous methods and techniques proposed in the literature (*Medimegh, Belaid & Werghi, 2015*; *Wang et al., 2008*; *Chou & Tseng, 2007*; *Garg, 2022*). These methodologies can be categorized into spatial domain, spectral domain, and transform domain based on the utilized embedding domain for the watermarking process.

Spatial domain techniques operate directly on the geometric properties of 3D objects, such as vertices' coordinates and mesh topology (*Zuliansyah et al., 2008*). One of the earliest techniques proposed in this domain was the vertex displacement technique. This technique modifies vertices' coordinates to embed the watermark. However, it is suffering from low robustness and low capacity. Other spatial domain techniques include the voxel-based method and face-based method, which operate on the voxel grid and the face normal of the object, respectively (*Sharma & Kumar, 2020*).

Spectral domain techniques operate on the spectral characteristics of the 3D object, such as the eigenvalues and eigenvectors of the Laplacian matrix (*Wu & Kobbelt, 2005*). These techniques have been shown to provide high robustness and high capacity but may suffer from low invisibility. Examples of spectral domain techniques include the frequency domain and Fourier domain embedding techniques (*Murotani & Sugihara, 2003*; *Abdallah, Ben Hamza & Bhattacharya, 2009*).

Transform domain techniques use both DWT and wavelet transformations to extract the needed coefficients of 3D objects (*Kanai, Date & Kishinami, 1998*; *Uccheddu, Corsini & Barni, 2004*). Such techniques are characterized by the trading-off property among invisibility and robustness processes and have been widely used in the literature. Examples of transform domain techniques include the wavelet-based embedding method and the DWT-based embedding method (*Kim et al., 2005*).

In a study by *Jani Anbarasi & Narendra (2017)*, a watermarking method for 3D meshes that focused on scalability and flexibility was presented. The proposed method used a spread-spectrum watermarking approach and was evaluated in terms of robustness against various security attacks, such as mesh simplification, scaling, smoothing, and noise addition. The results of *Jani Anbarasi & Narendra (2017)* showed that the suggested method was effective in providing robustness against such attacks. Another watermarking scheme was proposed by *Liang et al. (2020)*. Their proposed scheme is based on the quaternion Fourier transform (QFT) and uses a key-dependent approach to embed the

watermark into the model, making it more robust against attacks. The scheme was tested against well-known attacks, including mesh smoothing, scaling, and cropping. The findings showed the method's effectiveness in providing robustness against such attacks. Recently, *Qin, Sun & Wang (2015)* proposed a novel watermarking scheme based on the digital holography technique for 3D models. The proposed method was evaluated in terms of robustness against well-known attacks, including scaling, rotation, and translation. The results showed that the proposed method effectively provided robustness against such attacks.

*Yin et al. (2001)* and *Cayre & Macq (2003)* proposed spatial domain models that embed binary logos into 3D objects. These works achieve high invisibility but low robustness against geometric and signal-processing attacks. *Al-Saadi, Ghareeb & Elhadad (2021)* proposed a transform domain technique that embeds a binary logo into a 3D object by efficiently modifying the wavelet coefficients. This method achieves high robustness but medium invisibility. *Cui, Wang & Niu (2017)* proposed a technique based on shape signature and local feature points, which achieves high robustness and invisibility. However, the method requires the original 3D object to have well-defined feature points, which may not be available in all cases.

In a recent study, a 3D object watermarking scheme was proposed by *Huang (2023)* based on the combination of shape signature and local feature points. The shape signature represents the 3D object shape, while the local feature points capture the geometrical features of the object. The watermark is embedded, based on the shape signature, by modifying the positions of the local feature points based on the shape signature. The method achieves high robustness against well-known attacks, such as rotation and translation attacks, while maintaining high invisibility. However, the proposed technique requires the original 3D object to have well-defined feature points, which may not be available in all cases. In another study, *Abdallah, Ben Hamza & Bhattacharya (2009)* presented a method for 3D mesh watermarking that utilizes the curvature information of the mesh. The proposed technique decomposes the mesh into a set of sub-meshes and hides the watermark's information into a sub-mesh with the highest curvature. The watermarking process is achieved by modifying the vertex positions of the sub-mesh. The method achieves good robustness against well-known attacks while maintaining high invisibility. However, the proposed method has limited capacity and may not be suitable for larger watermarks.

*Kumar, Singh & Yadav (2020)* presented an extensive survey on multimedia and database watermarking, highlighting key trends and challenges in the field over recent years. Their work provided a foundational understanding of the diverse approaches to watermarking and their applicability across different media types. *Wan et al. (2022)* delve into the robustness of image watermarking techniques, presenting a comprehensive review of methods designed to ensure watermark integrity in the face of sophisticated attacks. This study underscores the importance of robustness as a critical attribute of effective watermarking schemes. Furthermore, *Kumar et al. (2023)* introduced an innovative entropy-based adaptive color image watermarking technique that operates within the YCbCr color space, showcasing the application of entropy measures to enhance watermark security and imperceptibility. Their approach exemplifies the ongoing evolution

**Table 1  Comparison of existing mesh-based and shape-based watermarking techniques.**

| Technique | Strengths | Weaknesses |
|---|---|---|
| Mesh-based geometry | High efficiency, robustness against common mesh processing | Limited resistance against geometric distortions |
| Mesh-based topology | High robustness against mesh editing operations | Limited resistance against geometric distortions |
| Shape-based volumetric | High resistance against geometric distortions | Limited efficiency |
| Shape-based feature | High invisibility, resistance against surface processing operations | Limited robustness against geometric distortions |

of watermarking methodologies to address emerging security needs. In *van Rensburg et al. (2023)*, introduced a watermarking method that leverages the homomorphic properties of the Paillier cryptosystem for high-capacity data hiding within the encrypted domain of 3D objects. This method distinguished itself by enabling the embedding of multiple secret messages without the size expansion of the original file.

Table 1 summarizes the strengths and weaknesses of the existing mesh-based and shape-based watermarking techniques. While mesh-based techniques are typically more efficient and provide better robustness against common mesh processing operations, shape-based techniques offer better resistance against geometric distortions and better invisibility. Overall, the choice of the watermarking technique depends on the specific application requirements and characteristics of the 3D object.

In comparison, our suggested solution conceals grayscale images as the watermark and is based on the DWT of the 3D object vertices. While not requiring particular feature points or additional embedding domains, the approach provides excellent robustness and invisibility. Numerous 3D object watermarking approaches have been proposed, but more effective and reliable techniques that can offer a higher level of security and invisibility are still required. These drawbacks are addressed by the suggested DWT-based watermarking methodology, which also attempts to safeguard the IP rights of 3D objects better.

## METHODOLOGY

As indicated in Fig. 1, the proposed watermarking model involves two main phases with several entailed steps. At the beginning, the normalization step normalizes the secret grayscale image and the original 3D object. Further, in the preprocessing step of our proposed watermarking model, DWT is strategically applied to sets of three vertices at a time. This specific approach is chosen based on the geometric structure of 3D objects, where vertices define the object's shape and spatial characteristics. Typically, a 3D object is represented as a mesh composed of numerous polygons, often triangles, which are themselves defined by three vertices. By applying DWT to every set of three vertices corresponding to a polygon, we can more accurately capture and utilize the local geometric features of the object for watermark embedding. Simultaneously, the watermark secret image undergoes reshaping and encryption processes. The watermarking integration phase is then performed by utilizing both the coefficients of the 3D object's vertices and the secret

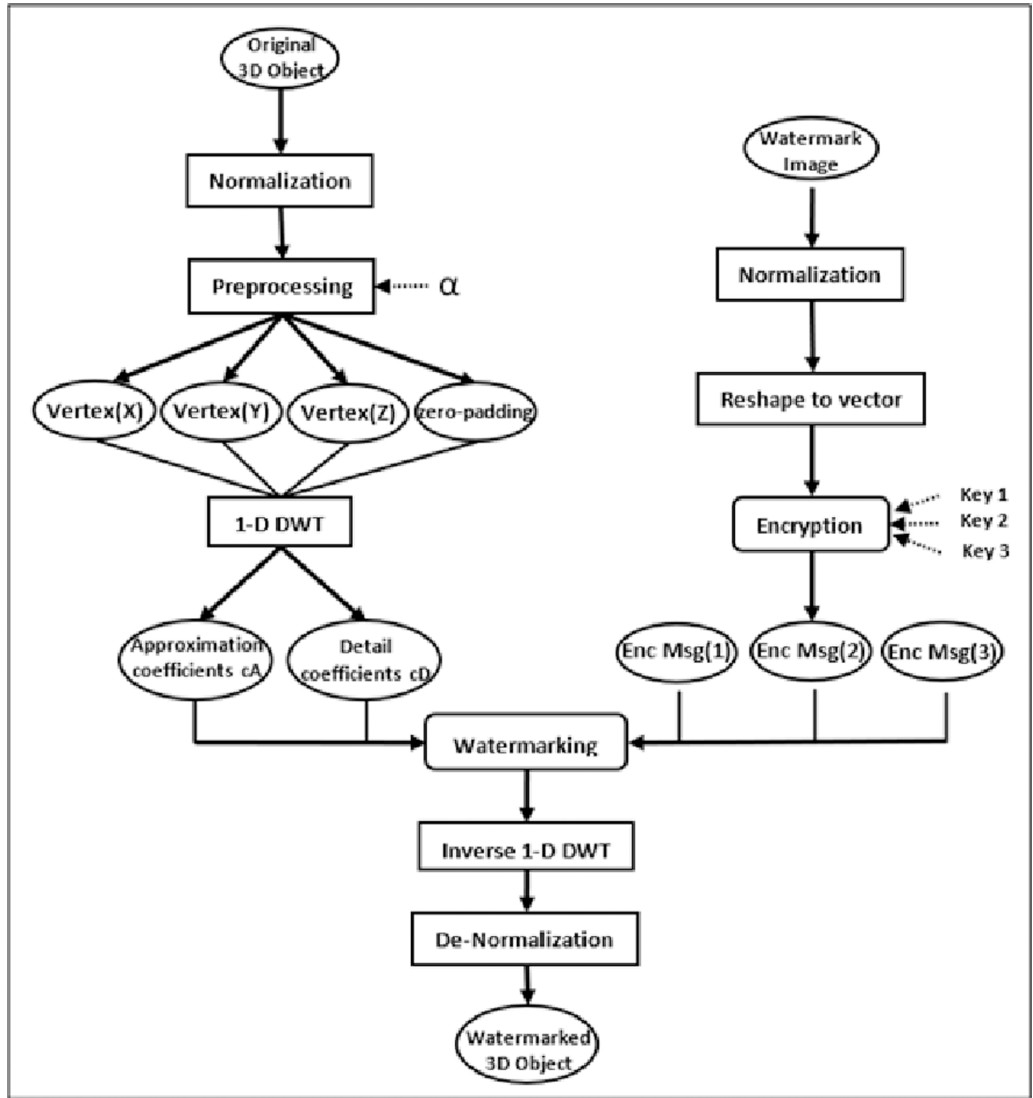

**Figure 1** The conceptual framework for the proposed watermarking method.

encrypted image pixels. Finally, the modified vertices are then subjected to inverse DWT and de-normalization to generate the watermarked 3D object.

## Watermark integration phase

The integration phase that embeds the watermark can be divided into three main steps: normalization, preprocessing, and 1-D DWT. The normalization step is the process that is usually used to change the range of data value dynamically; the 3D object or polygonal mesh object is converted to a standard format to ensure that the watermark is embedded uniformly across different objects. In the normalization process, transforming the range of values of the data object $\mathrm{Obj} : \left\{ X \subseteq \mathbb{R}^d \right\} \rightarrow \{\mathrm{Min}, \cdots, \mathrm{Max}\}$ to a new range of values $[\mathrm{Min}_{\mathrm{New}}, \mathrm{Max}_{\mathrm{New}}]$ in a new data object $\mathrm{Obj}_{\mathrm{New}} : \left\{ X' \subseteq \mathbb{R}^d \right\} \rightarrow \{\mathrm{Min}_{\mathrm{New}}, \cdots, \mathrm{Max}_{\mathrm{New}}\}$. This

is done using a linear normalization formula, which alters the original range of values [Min, Max]. Equation (1) presents a general formula for linear normalization, which is widely applicable across various data ranges that adjust each value x in the original range to a new value x′ in the new range using the equation:

$$\text{Obj}_{\text{New}} = (\text{Obj} - \text{Min}) \frac{\text{Max}_{\text{New}} - \text{Min}_{\text{New}}}{\text{Max} - \text{Min}} + \text{Min}_{\text{New}}. \tag{1}$$

This equation is a universal approach that adjusts data values from an original range [Min, Max] to a new specified range [Min$_{\text{New}}$, Max$_{\text{New}}$].

The suggested watermarking approach normalizes the original 3D object vertices and the secret grayscale image data, transforming the data value range into an intensity range of 0 to 1. This process ensures consistency in the range of the input data, which facilitates the watermark embedding. Specifically, the normalization is accomplished by using the following formula shown in Eq. (2):

$$\text{Obj}_{\text{New}} = \frac{(\text{Obj} - \text{Min})}{\text{Max} - \text{Min}}. \tag{2}$$

In the preprocessing step, a parameter $\alpha$ is used to adjust the normalized vertices of the original 3D object. This step is necessary to prevent overflow caused by saturated vertex values during the embedding process. The value of $\alpha$ is a small positive real number satisfying the condition $0 < \alpha < 1$. The adjustment process described by Eq. (3) is applied individually to each dimension $(x, y, z)$ of a vertex. Specifically, if any dimension of a vertex has a value of 0, that dimension is adjusted to $\alpha$. Similarly, if any dimension has a value of 1, it is adjusted to $1 - \alpha$. This ensures that each dimension of every vertex is kept within a permissible range that prevents overflow during the embedding process while still maintaining the geometric integrity of the 3D object.

$$\text{3D Obj(Vertex)} = \begin{cases} \alpha, & \text{if Vertex dimension value} = 0 \\ 1 - \alpha, & \text{if Vertex dimension value} = 1 \end{cases}. \tag{3}$$

Since DWT is a mathematical tool used to analyze signals in a multi-resolution way, it decomposes a signal into a set of coefficients that represent the signal at different scales and locations. DWT is performed by convolving the signal with a set of filters called the wavelet filters. The wavelet filters are composed of a scaling filter and a wavelet filter. The signal's low-frequency components are examined using the scaling filter, whereas the high-frequency components are studied with the wavelet filter. Therefore, DWT returns the approximation coefficients vector cA and detail coefficients vector cD of the vector.

The 1-D DWT step starts with a 3D object Obj as a set of vertices L, whereas each vertex is defined as Vertex(X, Y, Z), two sets of coefficients are computed for each vertex: approximation coefficients cA, and detail coefficients cD. The approximation and detail coefficients from the input 3D object are obtained by convolving Obj with the scaling filter LoD, which is the lowpass decomposition filter, followed by dyadic decimation, resulting in the approximation coefficients. On the other hand, convolving Obj with the wavelet filter HiD, which is the high pass decomposition filter, followed by dyadic decimation, yields the detail coefficients as illustrated in Fig. 2. To deal with the three vertices effects (signal-end)

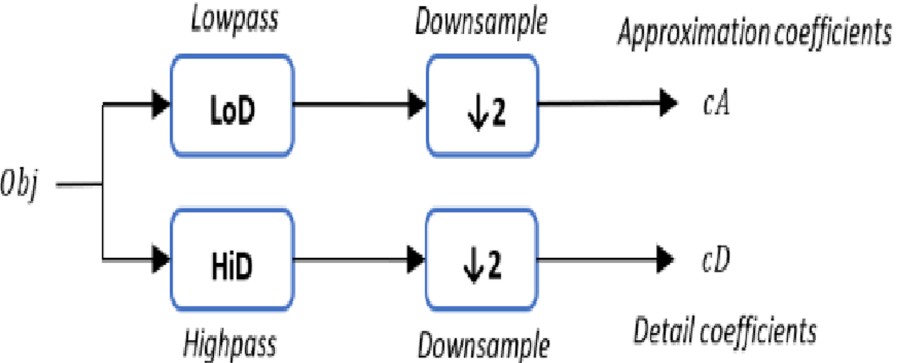

**Figure 2  The (1-D) discrete wavelet transform.**

resulting from a convolution-based stage, the possible option includes zero-padding. The length of each filter is equal to two coefficients.

The watermarking technique is conducted on the normalized coefficients of the 3D object vertices after the 1-D DWT stage. The normalization of the vertices helps in applying the 1-D DWT, which leads to the approximation coefficients within the range of [0, 2] and the detail coefficients within the range of [−1, 1]. As a result, we can build an equation system for watermarking the hidden grayscale picture in the modified areas of the 3D object coefficients. Therefore, the watermarking process is performed using Eqs. (4) and (5):

$$\text{3D obj}(\hat{C}) = \frac{2}{\beta}\left(\text{EncMsg}+i\right), \quad \frac{2i}{\beta} \leq \text{3D obj}(C) < \frac{2(i+1)}{\beta}. \tag{4}$$

$$i = \begin{cases} 0,1,2, \ldots(\beta-1), & C=\{cA_1, cA_2\} \\ -1,0,1,\ldots(\beta-3), & C=\{cD_1\} \end{cases}. \tag{5}$$

where 3D obj(C) refers to the original coefficient associated with a vertex in the 3D object before watermarking. 3D obj($\hat{C}$) represents the modified coefficient after the watermarking process has been applied to the vertices of the 3D object. EncMsg signifies the pixel value from the secret grayscale image that has been embedded into the 3D object. The parameter $\beta$ is used to indicate the total number of distinct intervals within which the coefficients, after normalization, are segmented. These intervals range either from [0, 2] for approximation coefficients or from [−1, 1] for detail coefficients, corresponding to DWT coefficients $cA_1$, $cA_2$ and $cD_1$. Each of these coefficients plays a crucial role in the DWT process, representing different aspects of the 3D object's information in the wavelet domain. Finally, the inverse of the *1-D* DWT and de-normalization are utilized to rebuild the watermarked *3D* object with the hidden embedded grayscale picture. Algorithm 1 describes the embedding process, including the mathematical connections written as pseudo-code.

The secret grayscale picture is encrypted using seed numbers created by a pseudorandom generator, which effectively shuffles the location of each pixel in the original image to

improve the security of the suggested watermarking approach. The encryption technique is done to the grayscale picture's rearranged vector to raise the scrambling's complexity even more. The location of the pixels is additionally changed using three secret keys to decrease the probability of it being decoded. This strategy enhances the watermarking model's security, making it immune against malicious attacks.

---

**Algorithm 1: Watermark Integrating Phase**

**Input:** The 3D object, Secret grayscale image, α, β, and Encryption keys

**Output:** The watermarked 3D object

1- Normalize the original 3D object and the secret grayscale image to $[0,1]$ using the Eq. (2) formula.

2- Reshape the Secret grayscale image into a vector.

3- Encrypt the secret grayscale image vector using a pseudorandom generator and three secret keys, resulting in a scrambled image vector EncMsg.

4- Preprocess the normalized 3D Object vertices using a small positive real number α as in the formula in Eq. (3).

5- Apply the 1-D DWT to each vertex of the 3D obj, resulting in a set of wavelet coefficients cA and cD.

6- Embed the vector EncMsg value within the $cA_1$, $cA_2$ and $cD_1$ coefficients values with the number of intervals β as shown in Eqs. (4) and (5).

7- Apply the inverse of 1-D DWT to each modified vertex of the 3D obj.

8- De-normalize the Watermarked 3D obj($\hat{C}$)

9- Output a 3D watermarked object.

---

## Watermark extraction phase

As illustrated in Fig. 3, the suggested approach for recovering the hidden image entails a series of phases that reverse the embedding process. First, the watermarked 3D object is normalized, and the vertices' 1-D DWT transform decomposition is calculated. Then, using the parameters and coefficients of the altered vertices 3D obj($\hat{C}$), the secret encrypted grayscale picture pixel is retrieved. Equations (6) and (7) are used for the extraction as follows:

$$\text{EncMsg} = \frac{\beta}{2}\left(3D\ \text{obj}(\hat{C}) - \frac{2i}{\beta}\right), \quad \frac{2i}{\beta} \leq 3D\ \text{obj}(\hat{C}) \leq \frac{2(i+1)}{\beta}. \tag{6}$$

$$i = \begin{cases} 0,1,2,3,\ldots(\beta-1), & \hat{C} = \{cA_1, cA_2\} \\ -2,-1,0,1,\ldots(\beta-3), & \hat{C} = \{cD_1\} \end{cases}. \tag{7}$$

The extraction process is performed blindly using the parameter β. Additionally, three keys are necessary to identify the genuine location of the encrypted pixels. Because the pixel values are normalized during the embedding process, they must be de-normalized before they can be returned to their original integer domain. Finally, the grayscale image is restored to the original image proportions using reshaping. In the extraction phase of our watermarking process, the hidden image is extracted three times to enhance the

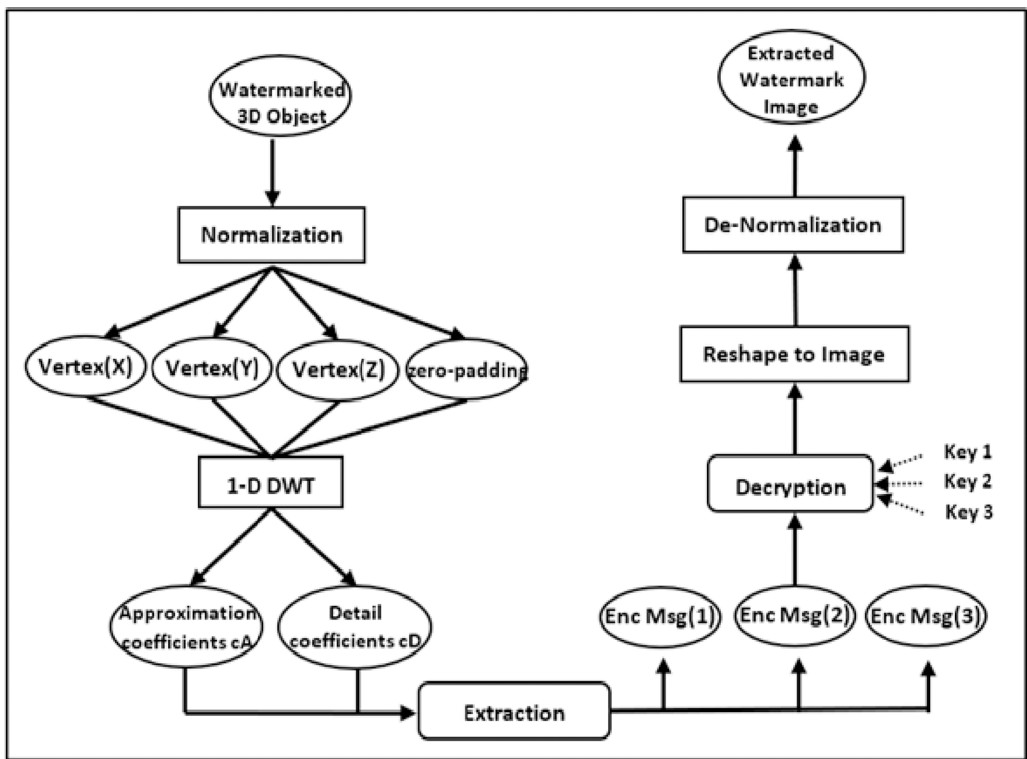

**Figure 3** The overall model of the extraction phase.

reliability and accuracy of watermark recovery. This methodological choice is grounded in the principle of redundancy, a well-established concept in error detection and correction techniques. By extracting the watermark multiple times, we introduce a layer of redundancy that allows for cross-verification among the extracted copies to identify and correct potential errors. This triple extraction process significantly reduces the probability of errors persisting in the final recovered image, thereby increasing the accuracy of the watermark recovery. In cases where discrepancies occur due to noise or distortions introduced during the watermarking or data transmission process, the majority vote principle is applied. For each pixel position, the value that appears in at least two of the three extractions is considered the correct value, thereby mitigating the impact of any singular extraction error. Algorithm 2 shows the precise steps for the extraction process.

The suggested extraction approach has the advantage of being blind, in which the original secret grayscale picture can be recovered without knowledge of the watermark or the embedding procedure. It is also robust, as it involves the use of three keys to identify the position of the original pixels in the normalized grayscale image, ensuring that the correct pixels are extracted. Additionally, the normalization and denormalization of the pixel values help increase the extraction process's complexity and security.

In our initial description of the watermark embedding and extraction processes, we assume that the size of the watermark image is commensurate with the number of vertices in the 3D object. However, practical applications often require embedding smaller-sized

---

**Algorithm 2:** Watermark Extraction Phase

**Input:** The watermarked 3D object, β, and Encryption keys

**Output**: The secret grayscale image

1- Normalize the watermarked 3D object to [0,1] using the Eq. (2) formula:

2- Apply the 1-D DWT to each vertex of the watermarked 3D obj, resulting in a set of wavelet coefficients cA and cD.

3- Extract the vector EncMsg value from the $cA_1$, $cA_2$ and $cD_1$ coefficients values as seen in equations Eqs. (6) and (7).

4- Reshape the secret grayscale image into the original dimensions.

5- Denormalize the secret grayscale image.

6- Output the targeted secret grayscale image.

---

watermark images into larger 3D models. To address this discrepancy and ensure clarity, we have refined our methodology to include a strategy for selecting vertices when embedding smaller watermark images and accurately identifying these vertices during the extraction process.

We employ a sampling strategy for embedding a smaller-sized watermark image that systematically selects a subset of vertices from the 3D object. This selection is based on a uniform sampling algorithm that ensures a representative distribution of vertices across the entire object. The algorithm divides the 3D object into regions proportional to the watermark image's dimensions, ensuring that each region contributes vertices for embedding the watermark information. This approach maintains the spatial integrity and uniformity of the watermark embedding process.

## EXPERIMENTAL RESULTS

### Implementation setup

We evaluated the performance of our watermarking technique using four common 3D objects. The dimensions of the secret grayscale images used as watermarks varied (597 × 349, 615 × 473, 1,119 × 453, and 1,728 × 823), aligning with the complexity and size of each 3D object. Our experiments were conducted on a system equipped with an Intel(R) Core(TM) i7-4700MQ CPU, a 2.40 GHz processor, and 16 GB of RAM, utilizing MATLAB version 9.9.0.1467703 (R2020b). Three distinct seeds (1987, 1989, and 1993) were used in the encryption process to ensure the robustness of our method.

### Capacity and payload analysis

When evaluating data hiding techniques, multiple parameters such as capacity and payload are considered, as seen in Eqs. (8) and (9). The greatest number of bits that may be buried in the vertices of a 3D object is referred to as its capacity. On the other hand, the actual payload is the fraction of presently implanted bits to the 3D object's capacity in bits. Table 2 shows the maximum capacity and actual payload for each 3D object, as well as the encoded hidden grayscale picture. The capacity in bits per vertex (bpv) and real payload percentage (%) may be determined using the following formulae and the number of vertices in the

**Table 2** The table presents the result of implementing 3D objects and the embedding capacity.

| Model | | | | |
|---|---|---|---|---|
| | Stanford Bunny | Horse | Cat Figurine | Angel |
| Vertices | 208,353 | 290,898 | 506,910 | 1,422,144 |
| Faces | 35,947 | 48,485 | 168,970 | 237,018 |
| Secret image size | 597 × 349 | 615 × 473 | 1,119 × 453 | 1,728 × 823 |
| Max Capacity | 5,000,472 | 6,981,552 | 12,165,840 | 34,131,456 |
| Actual Payload | 100 | 99.999 | 99.999 | 100 |

original 3D model (L):

$$\text{Capacity} = \frac{\text{Max(number of embedded pixels)} \times 8}{\text{number of vertices}} = \frac{L \times 8}{L} = 8\text{bpv}. \tag{8}$$

$$\text{Actual Payload} = \frac{\text{Secret image size in bits} \times 3 \times 100}{\text{3D object capacity in bits}}. \tag{9}$$

Table 2 presents the experimental results of the proposed method using four different 3D object models. The maximum embedding capacity of each object is also shown in terms of bits. It is observed that the larger the 3D object size, the higher the maximum capacity of the secret watermark image. The Stanford bunny, which is the smallest 3D object in the experiment, has a maximum capacity of 5,000,472 bits for a secret image size of 597 × 349 pixels. On the other hand, the angel 3D object, which is the largest in size, has a maximum capacity of 34,131,456 bits for a secret image size of 1,728 × 823 pixels.

Furthermore, the table also shows the actual payload achieved in terms of the percentage of the maximum capacity for each 3D object model. The proposed method achieved an actual payload of 100% for the Stanford bunny and angel models, indicating that the entire secret image was successfully embedded. For the horse and cat figurine models, the actual payload achieved was 99.999%, which is still a high percentage considering the complexity of the models. Overall, the findings show that the suggested technique is successful in terms of high embedding capacity and payload rate, especially for complicated 3D object models.

### Execution time performance

Figure 4 depicts the execution time of the watermarking and extraction operations for the four 3D objects. The findings reveal that the suggested technique has a modest computational complexity and requires a fair amount of time for embedding and extraction. For instance, the Stanford bunny 3D model, with the least number of vertices and faces, has the lowest execution time for both watermarking and extraction processes. On the other hand, the angel 3D model, with the highest number of vertices and faces, has the highest execution time for both processes. The average watermarking time for the Stanford

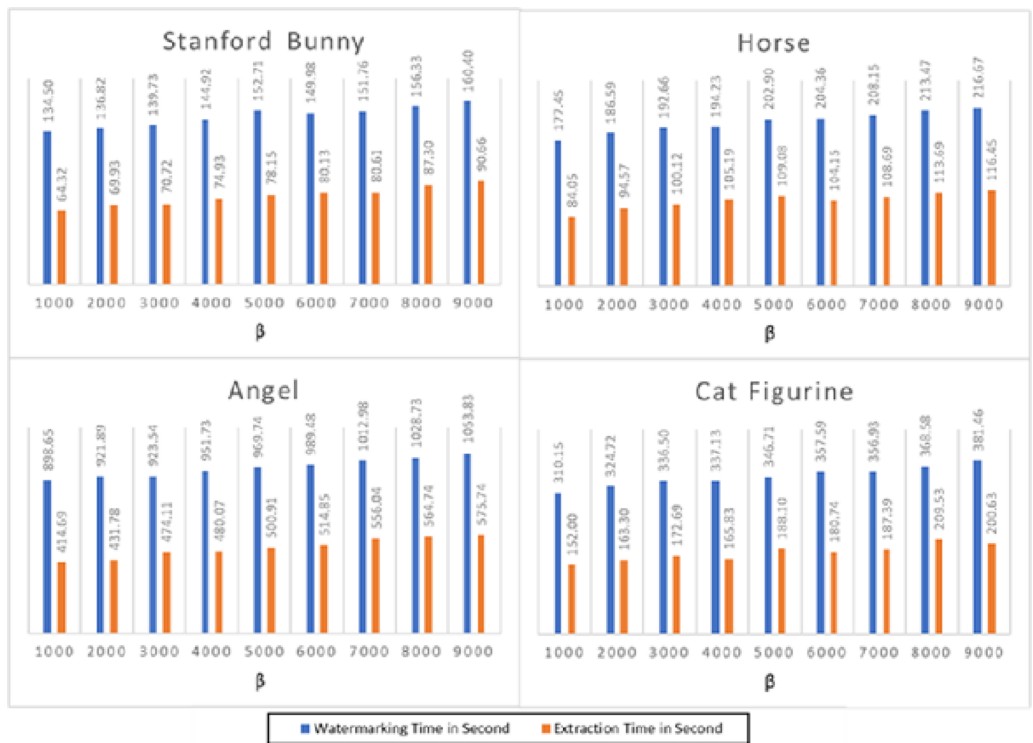

**Figure 4** The watermarking and extraction times performance of the proposed method.

bunny and horse models was approximately 140 s, while the average extraction time was approximately 70 s. For the angel and cat figurine models, the watermarking time was higher, averaging 980 s, and the extraction time was also longer, averaging 490 s. The proposed method shows a reasonable trade-off between capacity and computational complexity, making it a suitable option for practical applications.

Figure 4 presents the execution time results for the watermarking and extraction processes across four distinct 3D models: the Stanford bunny, horse, angel, and cat figurine, each tested at various values of β. The parameter β is instrumental in adjusting the watermark's strength; specifically, larger values of β correlate with the creation of stronger, more robust watermarks. This figure illustrates how changes in β affect the computational time required for embedding and extracting the watermark, providing insights into the trade-offs between watermark strength and processing efficiency. The results demonstrate that when the value grows, the watermarking time increases, but the extraction time stays almost constant for all models. This is because higher values of β require more computational resources to embed the watermark into the 3D models. The execution time for watermarking ranges from 134.50 s for the Stanford bunny at β = 1,000 to 160.40 s for the same model at β = 9,000. Similarly, the execution time for extraction ranges from 64.32 s for the Stanford bunny at β = 1,000 to 116.45 s for the horse at β = 9,000. Overall, the results demonstrate that the watermarking and extraction processes are feasible for large-scale 3D models, but the execution time is highly dependent on the value

of β. Therefore, the appropriate value of β should be selected based on the desired level of security and available computational resources. In addition, the results show that the execution time of the watermarking and extraction processes in 3D models is affected by the value of the parameter β and the complexity of the 3D model. Thus, it is crucial to carefully select the parameter β to achieve a balance between the watermarking strength and the execution time.

## Imperceptibility performance

The suggested method's imperceptibility performance is assessed by comparing the distances between the original 3D and watermarked objects. The evaluation revealed that a shorter distance provides a more invisible watermark. The performance of transparency is assessed by comparing the watermarked item and the extracted watermark. The greater the resemblance between the extracted and original watermarks, the clearer the watermarking approach. Several metrics are used to assess the invisibility performance of the proposed watermarking approach, including the Euclidean distance, Manhattan distance, cosine distance, and correlation distance. These metrics are used to compare the original 3D object, denoted as $u(u_x, u_y, u_z)$ with the watermarked object, denoted as $v(v_x, v_y, v_z)$. The imperceptibility of the watermark is assessed by measuring the difference between u and v using these metrics. Equations (10)–(13) provide more details on the invisibility performance terms:

**Euclidean distance**

$$\text{Euclidean dist}(u, v) = \sqrt{|u_x - v_x|^2 + |u_y - v_y|^2 + |u_z - v_z|^2}. \tag{10}$$

**Manhattan distance**

$$\text{Manhattan dist}(u, v) = |u_x - v_x| + |u_y - v_y| + |u_z - v_z|. \tag{11}$$

**Cosine distance**

$$\text{Cosine dist}(u, v) = 1 - \frac{u_x v_x + u_y v_y + u_z v_z}{\sqrt{|u_x|^2 + |u_y|^2 + |u_z|^2}\sqrt{|v_x|^2 + |v_y|^2 + |v_z|^2}}. \tag{12}$$

**The correlation distance**

$$\text{correlationdist}(u, v) = 1 - \frac{\text{Num}}{\text{Den}}, \text{ where}$$
$$\text{Num} = \left(\tfrac{1}{3}(-u_x - u_y - u_z) + u_x\right)\left(\tfrac{1}{3}(-v_x - v_y - v_z) + v_x\right) + \left(\tfrac{1}{3}(-u_x - u_y - u_z) + u_y\right)$$
$$\left(\tfrac{1}{3}(-v_x - v_y - v_z) + v_y\right) + \left(\tfrac{1}{3}(-u_x - u_y - u_z) + u_z\right)\left(\tfrac{1}{3}(-v_x - v_y - v_z) + v_z\right), \text{ and}$$

$$\text{Den} =: \alpha\beta. \tag{13}$$

Where $\alpha =$
$$\sqrt{\left|u_x + \tfrac{1}{3}(-u_x - u_y - u_z)\right|^2 + \left|u_y + \tfrac{1}{3}(-u_x - u_y - u_z)\right|^2 + \left|\tfrac{1}{3}(-u_x - u_y - u_z) + u_z\right|^2}, \text{ and}$$
$$\beta = \sqrt{\left|v_x + \tfrac{1}{3}(-v_x - v_y - v_z)\right|^2 + \left|v_y + \tfrac{1}{3}(-v_x - v_y - v_z)\right|^2 + \left|\tfrac{1}{3}(-v_x - v_y - v_z) + v_z\right|^2}.$$

In assessing the imperceptibility of the watermarking process, we employed Euclidean and Manhattan distances as metrics to evaluate the similarity between the original 3D models and their watermarked counterparts. Given the complexity of these models, characterized by a vast number of vertices, a straightforward comparison of distances

**Table 3  Euclidean distance for the invisibility performance.**

| β | Stanford Bunny | Horse | Angel | Cat Figurine |
|---|---|---|---|---|
| 1,000 | 52.02 | 36.79 | 54.17 | 56.10 |
| 2,000 | 59.25 | 41.24 | 57.18 | 62.60 |
| 3,000 | 61.73 | 42.82 | 58.10 | 64.62 |
| 4,000 | 62.97 | 43.67 | 58.55 | 65.61 |
| 5,000 | 63.71 | 44.09 | 58.83 | 66.16 |
| 6,000 | 64.16 | 44.40 | 59.04 | 66.55 |
| 7,000 | 64.46 | 44.67 | 59.17 | 66.82 |
| 8,000 | 64.73 | 44.85 | 59.27 | 67.05 |
| 9,000 | 64.91 | 44.97 | 59.35 | 67.19 |

between individual vertices would not sufficiently capture the overall impact of the watermarking on the model's geometry. To address this, we compute the Euclidean and Manhattan distances across all corresponding vertices between the original and watermarked 3D models. Specifically, for each vertex in the original model, we calculate the distance to its corresponding vertex in the watermarked model. The corresponding vertex is defined by its position within the model's geometric structure, ensuring a one-to-one match between vertices in the original and watermarked models. Once these distances are computed for all vertices, we then calculate the average Euclidean and Manhattan distances for the entire model. This averaging process allows us to consolidate the individual vertex distances into a single metric that reflects the overall geometric alteration introduced by the watermarking process. By reporting these average distances, we provide a comprehensive measure of the watermark's impact on the 3D model's geometry, ensuring that our evaluation captures the subtleties of the watermarking's effect on model imperceptibility.

Table 3 displays the Euclidean distance assessment findings for the proposed watermarking approach on four 3D models. The table displays the performance of the suggested approach for various values of β from 1,000 to 9,000. The Euclidean distance calculates the distance between the original 3D model and the watermarked model. The table shows that the Euclidean distances are quite modest, indicating that the suggested approach has acceptable imperceptibility performance and that the watermarked models are visually comparable to the original models. The values of the Euclidean distances increase slightly as β increases, indicating that increasing the value of β may slightly impact the imperceptibility performance. However, the values are still relatively small and do not exceed 65, which means the proposed method maintains good imperceptibility even at high values of β. The increase in Euclidean distance is relatively small, indicating that the proposed watermarking method is performing well in imperceptibility. For instance, in the case of the Stanford bunny object, the Euclidean distance values range from 52.02 for β = 1,000 to 64.91 for β = 9,000, which means that the watermarked object is still very similar to the original object even at high values of β. Overall, the Euclidean distance evaluation results indicate that the proposed watermarking method effectively maintains the imperceptibility of 3D models while embedding watermarks.

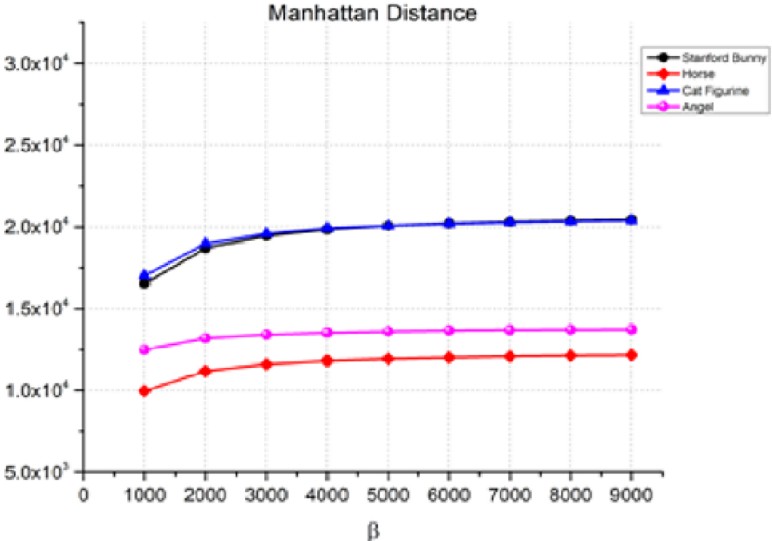

**Figure 5** **Manhattan distance for the invisibility performance.**

The Manhattan distance in Eq. (11) measures the absolute differences between the corresponding coordinates of two points in a plane. Figure 5 shows the Manhattan distance between the 3D object and the watermarked object for various values of β, which is the watermark strength. As β increases, the Manhattan distance also increases, indicating a decrease in the quality of the watermarked object. For instance, in the case of the Stanford bunny, the Manhattan distance increases from 1.65E+04 for β = 1,000 to 2.04E+04 for β = 9,000. Similarly, we can observe a similar trend for the horse, angel, and cat figurines. However, it is essential to note that the rate of rise in the Manhattan distance changes based on the item and the value of β. Therefore, the transparency performance remains relatively stable, as indicated by the low values of the distance measure between the watermarked 3D model and the original watermark.

Another measure used to assess the quality of the watermarking process is the cosine distance, Eq. (12), between the original 3D object and the watermarked item. The cosine distance is used to compare the similarity of two non-zero vectors in an inner product space. Figure 6 shows that the cosine distance is very small (close to zero) for all four objects at all values of β. This suggests that the proposed watermarking method is very effective in terms of imperceptibility when the cosine distance is used as the performance metric. For example, in the case of the Stanford bunny, the cosine distance is 0.00228 at β = 1,000, and it only increases to 0.00353 at β = 9,000. This means that the watermark is almost undetectable to the naked eye. Like the Manhattan distance, the cosine distance rises as the watermark strength increases.

Another statistic used to assess the quality of the watermarking process is the correlation distance between the original 3D object and the watermarked item. The correlation distance measures the linear relationship between two variables. The results in Fig. 7 show that the correlation distance increases with an increase in the watermark strength β. For instance,

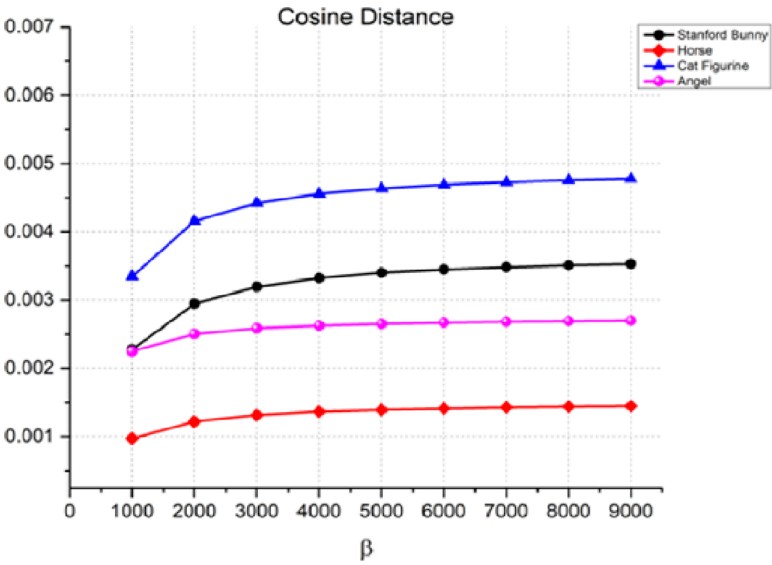

**Figure 6** Cosine distance for the invisibility performance of the proposed method.

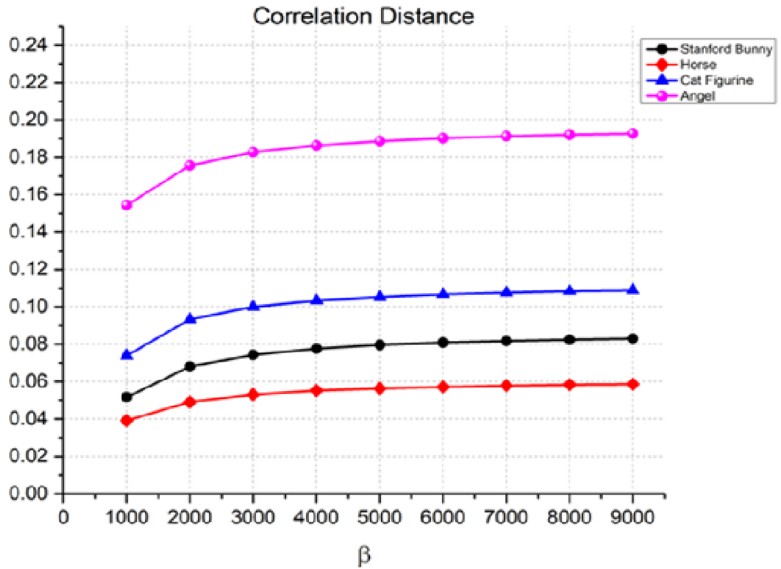

**Figure 7** Correlation distance for the invisibility performance of the proposed method.

in the case of the angel, the correlation distance increases from 0.1544 for β = 1,000 to 0.1089 for β = 9,000. This indicates that the watermark is more perceptible as β increases. Similarly, we can observe a similar trend for the Stanford bunny, horse, and cat figurine. However, it is worth noting that the rate of increase in the correlation distance is relatively slow compared to the other metrics, especially for the horse and cat figurine.

In our discussion of the imperceptibility of the watermarking process, we reported correlation distance values ranging from 0.1544 to 0.1089. It is crucial to understand the

context and scale of these values within the domain of digital watermarking and 3D model analysis. In the realm of 3D object watermarking, correlation distances are used to quantify the linear relationship between the geometric features of the original and watermarked models. Lower values indicate a higher degree of similarity, hence a more imperceptible watermarking effect.

Overall, the results indicated that as the watermark strength β increases, the quality of the watermarked object decreases. The rise in Manhattan, cosine, and correlation distances between the original 3D object and the watermarked object demonstrates this. The rate of increase in the distance metrics varies depending on the object and the value of β. Therefore, to achieve optimal watermarking performance, a balance between the watermark strength and the quality of the watermarked object must be achieved. However, the results also indicate that there is a trade-off between imperceptibility and transparency performance, with higher embedding strengths resulting in lower imperceptibility. Additionally, the relatively slow increase in the correlation distance compared to the other two metrics suggests that the correlation distance may be a better metric for evaluating the imperceptibility of the watermarking process. As a result, it is advised that while utilizing this watermarking approach, the Cosine distance be used as the performance parameter to guarantee that the watermark is nearly undetectable to the human eye.

## Structural similarity index evaluation of extracted image

The structural similarity index (SSIM) is a popular picture quality statistic for determining the similarity of two images. The SSIM metric is based on the idea that the human visual system is very sensitive to structural information in images and aims to capture this sensitivity. The SSIM metric, which ranges from 0 to 1, assesses the structural similarity between the derived secret grayscale image and the original secret grayscale image. A rating of 1 shows that the two photos are perfectly comparable. The definition of the SSIM metric is designed to measure the similarity between two images. SSIM is calculated as the following formula shown in Eq. (14):

$$SSIM(x,y) = \frac{(2\mu_x\mu_y + c_1)(2\sigma_{xy} + c_2)}{(\mu_x^2 + \mu_y^2 + c_1)(\sigma_x^2 + \sigma_y^2 + c_2)}. \tag{14}$$

Where $x$ and $y$ are the original and watermarked images, respectively. $\mu_x$ and $\mu_y$ are the average pixel values. $\sigma_x^2$ and $\sigma_y^2$ are the variances. $\sigma_{xy}$ is the covariance between $x$ and $y$. $c_1$ and $c_2$ are constants used to stabilize the division with a weak denominator.

In this case, the extracted secret grayscale image has an SSIM of 1 for all testing models, which means that it is perfectly like the original secret grayscale image. The high SSIM value also illustrates the proposed algorithm's ability to retain the quality of the watermarked 3D models while assuring that the secret image can be recovered consistently from the watermarked 3D model. This shows that the suggested watermarking approach can retrieve the secret image with high fidelity while generating little distortion. This is a significant finding since it demonstrates that the suggested approach can embed the secret picture efficiently and remove it with no loss of quality. Overall, the high SSIM values indicate

**Table 4  Robustness test results.**

| Attack test | | Extracted secret image SSIM |
|---|---|---|
| Scaling | uniform scaling by 2 | 0.64 |
| | uniform scaling by 3 | 0.64 |
| | uniform scaling by 4 | 0.64 |
| Translation | XYZ translation by $-1$ | 0.56 |
| | XYZ translation by 0.5 | 0.59 |
| | XYZ translation by 1 | 0.52 |
| Rotation | angle 90 | 0.63 |
| | angle 180 | 0.61 |
| | angle 270 | 0.67 |

that the proposed method is a promising technique for watermarking 3D models while maintaining their visual quality.

## Robustness against attacks

To thoroughly evaluate the robustness of our proposed watermarking method, we conducted a series of tests focusing on the resilience of the embedded watermark against common attacks such as rotation, scaling, and translation. These attacks simulate potential alterations a watermarked 3D object might undergo during its lifecycle, making their consideration crucial for assessing the practicality and security of watermarking techniques.

Our robustness tests were carried out using a 3D bunny model, chosen for its standard use in 3D graphics testing due to its complex geometry. The watermarking was performed with a parameter setting of $\beta = 500$ and a secret image of size $597 \times 349$ pixels embedded within the model. Following the embedding process, the watermarked 3D object was subjected to a series of transformations using MeshLab (v2016.12), an open-source system renowned for its comprehensive 3D processing capabilities. The transformations included 3D rotation, scaling, and translation, each designed to challenge the watermark's integrity and retrieval capabilities.

The ability to accurately retrieve the embedded secret image post-transformation is quantitatively assessed using the SSIM, a metric that measures the similarity between two images. Table 4 presents the SSIM values obtained for each attack, providing insights into the watermark's resilience. The experimental results indicate that the proposed watermarking technique maintains a high fidelity level in retrieving the secret image, even after applying various geometric transformations. Notably, the SSIM values remain significantly high across all tests, indicating that the embedded watermark is largely unaffected by rotation, scaling, and translation attacks.

## Comparative analysis

This section compares the proposed watermarking method against other existing techniques to highlight its validity and efficiency. The comparative study primarily focuses on several key aspects: the type of cover media used, the nature of the watermark sequence, the embedding space, the domain of operation, the capacity in terms of bits per pixel (bpp)

**Table 5  Comparison of recent schemes.**

| Scheme | Cover media | Watermark sequence | Domain | Capacity | Is blind? |
|---|---|---|---|---|---|
| *Delmotte et al. (2019)* | 3D printed object | Binary bits | Layer thickness | >64 bits | Yes |
| *Jiang et al. (2017)* | 3D object | Binary bits | Encrypted domain | 0.3692 bpv | Yes |
| *Cayre & Macq (2003)* | 3D object | Binary bits | Spatial | 0.8772 bpv | Yes |
| *Wu & Cheung (2006)* | 3D object | 2D binary image | Spatial | 0.9969 bpv | No |
| *Khalil, Elhadad & Ghareeb (2020)* | 3D object | Grayscale image | Spatial | 2.6667 bpv | Yes |
| The Proposed Model | 3D Object | Grayscale image | DWT | 8 bpv | Yes |

or bits per vertex (bpv), and whether the method supports blind extraction. Such a comparison is crucial for validating the superiority of the proposed method in terms of capacity and applicability across various media. Prior works, as reported in reference *Delmotte et al. (2019)* have ventured into watermarking 3D printed objects, while studies in *Khalil, Elhadad & Ghareeb (2020)*; *Jiang et al. (2017)*, *Cayre & Macq (2003)*, and *Wu & Cheung (2006)* have explored watermarking 3D objects with various watermark sequences as illustrated in  Table 5.

Among these methods, the capacities are notably distinguished by the bpp for image-based techniques and bpv for 3D object watermarking. Our proposed 3D objects watermarking technique stands out for its high-capacity embedding, measured in bpv, showcasing an advantageous characteristic over other schemes, particularly in the context of copyright protection where capacity and imperceptibility are paramount.

Our method, employing a grayscale image watermark embedded within a 3D object's DWT domain, achieves a substantial capacity of 8 bpv, significantly higher than the capacities reported in the related works. This capacity, coupled with the method's support for blind extraction, underscores the proposed technique's innovative approach to high-capacity and imperceptible 3D object watermarking.

## CONCLUSION

This work presents a unique technique for watermarking 3D objects based on the vertices' DWT features. We insert a secret grayscale image three times using the coefficients of the vertices and encrypted image pixels. The extraction procedure is blind and retrieves the hidden image by reversing the embedding processes. Our method's performance is evaluated using various distance metrics, which demonstrate its superiority in terms of execution time and imperceptibility. Four separate distance measurements assessed the performance of the proposed model on four different 3D objects. The outcomes showed that the suggested technique produced great imperceptibility and transparency with little distortion. An SSIM of 1 for all testing models revealed that the extracted hidden grayscale image and the original watermark were exactly matched. Thus, the proposed advanced model can offer a reliable and efficient method of copyright protection for 3D and polygonal mesh objects. Future work could focus on developing algorithms that intelligently select subsets of nodes based on the geometric and topological properties of

the 3D models. This would involve analyzing the model's structure to identify nodes that best balance watermark visibility and distortion minimization. Additionally, exploring different strategies for ordering these nodes could provide insights into how the spatial arrangement of embedded watermarks affects their detectability and resilience to various attacks.

### Funding
This work was supported by Princess Nourah bint Abdulrahman University Researchers Supporting Project Number (PNURSP2024R442), Princess Nourah bint Abdulrahman University, Riyadh, Saudi Arabia. The funders had no role in study design, data collection and analysis, decision to publish, or preparation of the manuscript.

### Grant Disclosures
The following grant information was disclosed by the authors:
Princess Nourah bint Abdulrahman University Researchers Supporting Project, Princess Nourah bint Abdulrahman University, Riyadh, Saudi Arabia:  PNURSP2024R442.

### Competing Interests
The authors declare there are no competing interests.

### Author Contributions
- Sarah M. Alhammad analyzed the data, authored or reviewed drafts of the article, and approved the final draft.
- Nada Ahmed analyzed the data, authored or reviewed drafts of the article, and approved the final draft.
- Safia Abbas conceived and designed the experiments, analyzed the data, prepared figures and/or tables, authored or reviewed drafts of the article, and approved the final draft.
- Hussein Abulkasim conceived and designed the experiments, performed the experiments, prepared figures and/or tables, authored or reviewed drafts of the article, and approved the final draft.
- Ahmed Elhadad conceived and designed the experiments, performed the experiments, performed the computation work, prepared figures and/or tables, authored or reviewed drafts of the article, and approved the final draft.

### Data Availability
   The four 3D objects used and the code of the model are available in the Supplemental Files.

### Supplemental Information
Supplemental information for this article can be found online at http://dx.doi.org/10.7717/peerj-cs.2020#supplemental-information.

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
