# Peer review of "Robust 3D object watermarking scheme using shape features for copyright protection"

_PeerJ Computer Science, doi:10.7717/peerj-cs.2020_

## Round 0.1 · original submission · Major Revisions

I have received reviews of your manuscript from scholars who are experts on the cited topic. They find the topic very interesting; however, several concerns must be addressed regarding experimental results, technical presentation, discussions, and comparisons with current approaches. These issues require a major revision. Please refer to the reviewers’ comments listed at the end of this letter, and you will see that they are advising that you revise your manuscript. If you are prepared to undertake the work required, I would be pleased to reconsider my decision. Please submit a list of changes or a rebuttal against each point that is being raised when you submit your revised manuscript.

Thank you for considering PeerJ Computer Science for the publication of your research.

With kind regards,

**Language Note:** The review process has identified that the English language must be improved. PeerJ can provide language editing services - please contact us at [email protected] for pricing (be sure to provide your manuscript number and title). Alternatively, you should make your own arrangements to improve the language quality and provide details in your response letter. – PeerJ Staff

Reviewer 1 ·

Basic reporting

All the points suggested by the reviewers are positively addressed by the authors.

Experimental design

NA

Validity of the findings

NA

Additional comments

NA

·

Basic reporting

• The basic reporting of the manuscript is not clear and the English language used is not clear

Experimental design

• The experimental setup can be improved. The proposed method is not compared with the state-of-the-art/related works in terms of experimentation, results, and suitability. It is also suggested to work on the different categories of datasets.

Validity of the findings

• Manuscript claims that the proposed method provides excellent robustness and invisibility but fails to give justification. The manuscript analysis is more on mathematical formulas and theoretical explanation and fails to give good analysis results. Tabular and graphical analysis are missing in the manuscript.

Additional comments

• More recent papers should be reviewed. There are still some more profound works that are not considered in this work.
The writing of this manuscript needs further improvements. There are some grammatical mistakes.

Reviewer 3 ·

Basic reporting

This paper presents a novel 3D object watermarking method using discrete wavelet transformation (DWT) based on vertex characteristics. The model consists of integration and extraction phases. In integration, a new technique embeds a grayscale image thrice using encrypted pixels and vertex coefficients. Extraction phase recaptures the secret image. Tested on common 3D objects, the method shows high imperceptibility and transparency with low distortion. The extracted image matches the original watermark perfectly (SSIM=1). This work is interesting and well-organized. The work also adds value to the research field. Before moving forward with the acceptance, there is a need for some minor revisions, which are as follows:
A few of the grammatical mistakes were found in the whole draft of the article.
For examples:
In line 302, “for various values of.”
In line 303, “The value of is used”
In the abstract, use full terms, not abbreviations, e.g., discrete wavelet transformation (DWT).
In the conclusion, The authors should only use the abbreviation DWT, and there is no need to write the complete term discrete wavelet transform. A comprehensive review of all abbreviations is required.
In line 77, the authors mention that “Section 2 reviews the related work …”. However, all sections of the manuscript are not numbered.
Newer research can be referenced to align this work with current advancements, such as “van Rensburg, Bianca Jansen, et al. "3D Object Watermarking from Data Hiding in the Homomorphic Encrypted Domain." ACM Transactions on Multimedia Computing, Communications and Applications 19.5s (2023): 1-20.”
In equation 9, “Actul Payload” should be “Actual Payload”.

Experimental design

-

Validity of the findings

-

·

Basic reporting

This paper titled “Robust 3D Object Watermarking Scheme Using Shape Features for Copyright Protection” provides a 3D object watermarking method utilizing 1-D discrete wavelet transform of the normalized vertex coordinates. Three wavelet coefficients (cA1, cA2 and cD1) are computed for each vertex and a copy of the watermark gray scale image pixel is embedded in each of the 3 wavelet coefficients. The grayscale image to be used as watermark is also normalized to range [0, 1]. The range of wavelet coefficients are divided into  number of intervals. Wavelet coefficients are modified to embed the watermark depending upon the interval in which a wavelet coefficient falls. In the extraction phase, the watermark image is extracted using the inverse of the embedding process. Four common 3D objects (Stanford Bunny, Horse, Cat Figurine, and Angel), were used in this model as datasets. The experimental results demonstrated that the proposed method achieved high imperceptibility with low distortion in the extracted watermark.

The paper is found to be technically sound making it fit for publication. But technical presentation, particularly the methodology section needs some minor corrections. A few sentences of the basic reporting text also need minor corrections. Some sentences need elaboration as they are found to be ambiguous. The paper can be accepted after the corrections are done. The suggested modifications required are listed below:

1. Line 71-73: “The experiment used four 3D objects … as seen in Table 1”. But Table 1 contains Comparison of existing mesh-based and shape-based watermarking techniques.
2. Line 157-158: “Further, the preprocessing step implements the DWT on every three vertices.” Meaning of the line is not clear. It needs elaboration.
3. Line 169-175: What is the need of the statements including Equation (1) as the desired normalization is achieved directly using Equation (2). Algorithm 1 never used Equation (1).
4. Line 186: In Equation (3), what is meant by Vertex=0 and Vertex=1. If only one dimension, say x-dimension of the 3D Vertex(x, y, z) is 0, then the equation should be applied or not?
5. Line 211: In Equation (5) range of values of i, that is i=-2, -1, 0, 1, … (-3) C={cD1} seems to be incorrect as it violates allowable range [-1, 1] of detailed coefficients as stated in line 207. Putting the values i=-2 or i=-3 in Equation (4) will cross the range [-1, 1].
6. Line 212-215: The complex sentence “Where 3D obj(C) is the equivalent coefficient … and cD1 are the wavelet coefficients.” is hard to interpret and seems to be grammatically incorrect. It should be rewritten.
7. Line 241-242: “It should be mentioned that the hidden grayscale picture is extracted three times to ensure accuracy.” Explanation required regarding how embedding and extracting 3 times increases accuracy. How are the three copies used? If one copy can be inaccurate, so be the 2nd and 3rd copy. It raises doubt regarding robustness claim.
8. Line 301-304. The two sentences included in the lines are incomplete.
9. Line 331 and 333: The Euclidean distance and Manhattan distance is distance between two vertices. The watermark image and extracted watermark image contains so many vertices, then how is the Euclidean distance computed and reported for the experiments performed. Is average of the distances used?
10. Line 389: The reported correlation distance 0.1544 to 0.1089 is very very low. It raises doubt regarding imperceptibility claim.
11. Line 408: The correlation measure could have been computed along with SSIM to show the robustness of the extracted watermark. Definition of SSIM metric can be included. Moreover, attacks are not considered for evaluating the robustness of the method. Is it justified to claim the proposed method to be robust without performing experiments with attacks?
12.The sizes of the images embedded as watermarks are considered equal to the number of vertices in the 3D objects. If a smaller sized watermark image is embedded, then how will the vertices be selected for embedding and identified for extraction?

Experimental design

Attacks are not considered for experimentation. Comments included in the previous section.

Validity of the findings

Findings are found to be valid based on reported experimental results. Comments included in the previous section.

Additional comments

Selecting specific subset of nodes and ordering them in some order are important factors for watermark embedding. Also section of starting node is equally important. These factors may be considered for future works.

---

## Round 0.2 · accepted · Accept

I am pleased to inform you that your work has now been accepted for publication in PeerJ Computer Science.

Please be advised that you are not permitted to add or remove authors or references post-acceptance, regardless of the reviewers' request(s).

Thank you for submitting your work to this journal. On behalf of the Editors of PeerJ Computer Science, we look forward to your continued contributions to the Journal.

With kind regards,

Reviewer 3 ·

Basic reporting

The authors have addressed all my issues in the revised manuscript. Further, I have no comments. The manuscript is good, and I recommend for publishing.

Experimental design

-

Validity of the findings

-

·

Basic reporting

After proper addressing of the issues raised the article is found to be understandable and technically sound.

Experimental design

No comment.

Validity of the findings

No comment.